# LEARNING TO DECOMPOSE COMPOUND QUESTIONS WITH REINFORCEMENT LEARNING

## ABSTRACT

As for knowledge-based question answering, a fundamental problem is to relax the assumption of answerable questions from simple questions to compound questions. Traditional approaches firstly detect topic entity mentioned in questions, then traverse the knowledge graph to find relations as a multi-hop path to answers, while we propose a novel approach to leverage simple-question answerers to answer compound questions. Our model consists of two parts: (i) a novel learning-to-decompose agent that learns a policy to decompose a compound question into simple questions and (ii) three independent simple-question answerers that classify the corresponding relations for each simple question. Experiments demonstrate that our model learns complex rules of compositionality as stochastic policy, which benefits simple neural networks to achieve state-of-the-art results on WebQuestions and MetaQA. We analyze the interpretable decomposition process as well as generated partitions.

## 1 INTRODUCTION

Knowledge-Based Question Answering (KBQA) is one of the most interesting approaches of answering a question, which bridges a curated knowledge base of tremendous facts to answerable questions. With question answering as a user-friendly interface, users can easily query a knowledge base through natural language, i.e., in their own words. In the past few years, many systems (Berant et al., 2013; Bao et al., 2014; Yih et al., 2015; Dong et al., 2015; Zhang et al., 2017; Hao et al., 2017) have achieved remarkable improvements in various datasets, such as WebQuestions (Berant et al., 2013), SimpleQuestions (Bordes et al., 2015) and MetaQA (Zhang et al., 2017).

However, most of them (Yih et al., 2014; Bordes et al., 2015; Dai et al., 2016; Yin et al., 2016; Yu et al., 2017) assume that only simple questions are answerable. **Simple questions** are questions that have only one relation from the topic entity to unknown tail entities (answers, usually substituted by an interrogative word) while **compound questions** are questions that have multiple[1] relations. For example, "Who are the daughters of Barack Obama?" is a simple question and "Who is the mother of the daughters of Barack Obama?" is a compound question which can be decomposed into two simple questions.

In this paper, we aim to relax the assumption of answerable questions from simple questions to compound questions. Figure 1 illustrates the process of answering compound questions. Intuitively, to answer a compound question, traditional approaches firstly detect topic entity mentioned in the question, as the starting point for traversing the knowledge graph, then find a chain of multiple ($\leq 3$) relations as a multi-hop[2] path to golden answers.

We propose a learning-to-decompose agent which assists simple-question answerers to solve compound questions directly. Our agent learns a policy for decomposing compound question into simple ones in a meaningful way, guided by the feedback from the downstream simple-question answerers. The goal of the agent is to produce partitions and compute the compositional structure of questions

---

[1] We assume that the number of corresponding relations is at most three.

[2] We are aware of the term *multi-hop question* in the literature. We argue that *compound question* is a better fit for the context of KBQA since *multi-hop* characterizes a path, not a question. As for document-based QA, *multi-hop* also refers to routing over multiple evidence to answers.

Q: the films that share actors with the film *Catch Me If You Can* are written by who

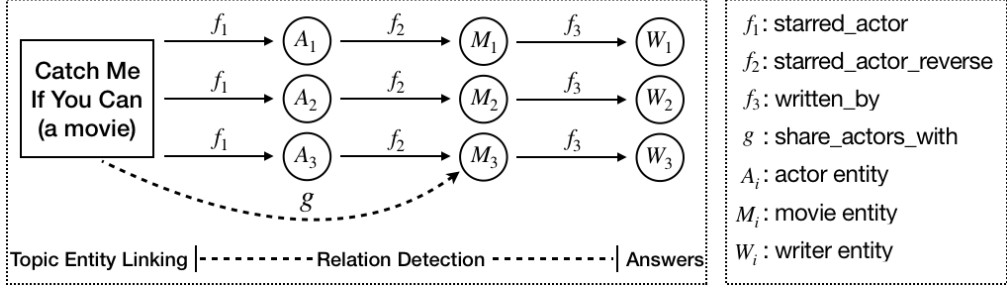

Figure 1: An example of answering compound questions. Given a question $Q$, we first identify the topic entity $e$ with entity linking. By relation detection, a movie-to-actor relation $f_1$, an actor-to-movie relation $f_2$ and a movie-to-writer relation $f_3$ forms a path to the answers $W_i$. Note that each relation $f_i$ corresponds to a part of the question. If we decomposes the question in a different way, we may find a movie-to-movie relation $g$ as a shortcut, and $g(e) = f_2(f_1(e)) = (f_2 \circ f_1)(e)$ holds. Our model discovered such composite rules. See section 4 for further discussion.

with maximum information utilization. The intuition is that encouraging the model to learn structural compositions of compound questions will bias the model toward better generalizations about how the meaning of a question is encoded in terms of compositional structures on sequences of words, leading to better performance on downstream question answering tasks.

We demonstrate that our agent captures the semantics of compound questions and generate interpretable decomposition. Experimental results show that our novel approach achieves state-of-the-art performance in two challenging datasets (WebQuestions and MetaQA), without re-designing complex neural networks to answer compound questions.

## 2 RELATED WORK

### 2.1 KNOWLEDGE-BASED QUESTION ANSWERING

For combinational generalization (Battaglia et al., 2018) on the search space of knowledge graph, many approaches (Yih et al., 2014; Yin et al., 2016; Zhang et al., 2017) tackle KBQA in a tandem manner, i.e., topic entity linking followed by relation detection. An important line of research focused on directly parsing the semantics of natural language questions to structured queries (Cai & Yates, 2013; Kwiatkowski et al., 2013; Yao & Van Durme, 2014; Yao et al., 2014; Bao et al., 2014; Yih et al., 2014; 2015). An intermediate meaning representation or logical form is generated for query construction. It often requires pre-defined rules or grammars (Berant et al., 2013) based on hand-crafted features.

By contrast, another line of research puts more emphasis on representing natural language questions instead of constructing knowledge graph queries. Employing CNNs (Dong et al., 2015; Yin et al., 2016) or RNNs (Dai et al., 2016; Yu et al., 2017), variable-length questions are compressed into their corresponding fix-length vector. Most approaches in this line focus on solving *simple questions* because of the limited expression power of fix-length vector, consistent with observations (Sutskever et al., 2014; Bahdanau et al., 2015) in Seq2Seq task such as Neural Machine Translation.

Closely related to the second line of research, our proposed model learns to decompose *compound question* into simple questions, which eases the burden of learning vector representations for compound question. Once the decomposition process is completed, a simple-question answerer directly decodes the vector representation of simple questions to an inference chain of relations with the desired order, which resolves the bottleneck of KBQA.

## 2.2 REINFORCEMENT LEARNING FOR NATURAL LANGUAGE UNDERSTANDING

Many reinforcement learning approaches learn sentence representations in a bottom-up manner. Yogatama et al. (2017) learn tree structures for the order of composing words into sentences using reinforcement learning with Tree-LSTM (Tai et al., 2015; Zhu et al., 2015), while Zhang et al. (2018) employ REINFORCE (Williams, 1992) to select useful words sequentially. Either in tree structure or sequence, the vector representation is built up from the words, which benefits the downstream natural language processing task such as text classification (Socher et al., 2013) and natural language inference (Bowman et al., 2015). By contrast, from the top down, our proposed model learns to decompose compound questions into simple questions, which helps to tackle the bottleneck of KBQA piece by piece. See section 3 for more details.

Natural question understanding has attracted the attention of different communities. Iyyer et al. (2017) introduce SequentialQA task that requires to parse the text to SQL which locates table cells as answers. The questions in SequentialQA are decomposed from selected questions of WikiTable-Questions dataset (Pasupat & Liang, 2015) by crowdsourced workers while we train an agent to decompose questions using reinforcement learning. Talmor & Berant (2018) propose a ComplexWe-bQuestion dataset that contains questions with compositional semantics while Bao et al. (2016) collects a dataset called ComplexQuestions focusing on multi-constrained knowledge-based question answering.

The closest idea to our work is Talmor & Berant (2018) which adopts a pointer network to decompose questions and a reading comprehension model to retrieve answers from the Web. The main difference is that they leverage explicit supervisions to guide the pointer network to correctly decompose complex web questions based on human logic (e.g., conjunction or composition) while we allow the learning-to-decompose agent to discover good partition strategies that benefit downstream task. Note that it is not necessarily consistent with human intuition or linguistic knowledge.

## 2.3 DEEP SEMANTIC ROLE LABELING

Without heavy feature engineering, semantic role labeling based on deep neural networks (Collobert et al., 2011) focus on capturing dependencies between predicates and arguments by learning to label semantic roles for each word. Zhou & Xu (2015) build an end-to-end system which takes only original text information as input features, showing that deep neural networks can outperform traditional approaches by a large margin without using any syntactic knowledge. Marcheggiani et al. (2017) improve the role classifier by incorporating vector representations of both input sentences and predicates. Tan et al. (2018) handle structural information and long-range dependencies with self-attention mechanism.

This line of work concentrates on improving role classifier. It still requires rich supervisions for training the role classifier at the token level. Our approach also requires to label an action for each word, which is similar to role labeling. However, we train our approach at the sentence level which omits word-by-word annotations. Our learning-to-decompose agent generates such annotations on the fly by exploring the search space of strategies and increases the probability of good annotations according to the feedback.

## 3 MODEL

Figure 2 illustrates an overview of our model and the data flow. Our model consists of two parts: a learning-to-decompose agent that decomposes each input question into at most three partitions and three identical simple-question answers that map each partition to its corresponding relation independently. We refer to the learning-to-decompose agent as *the agent* and three simple-question answerers as *the answerers* in the rest of our paper for simplicity.

## 3.1 LEARNING-TO-DECOMPOSE AGENT

Our main idea is to best divide an input question into at most three partitions which each partition contains the necessary information for the downstream simple-question answerer. Given an input

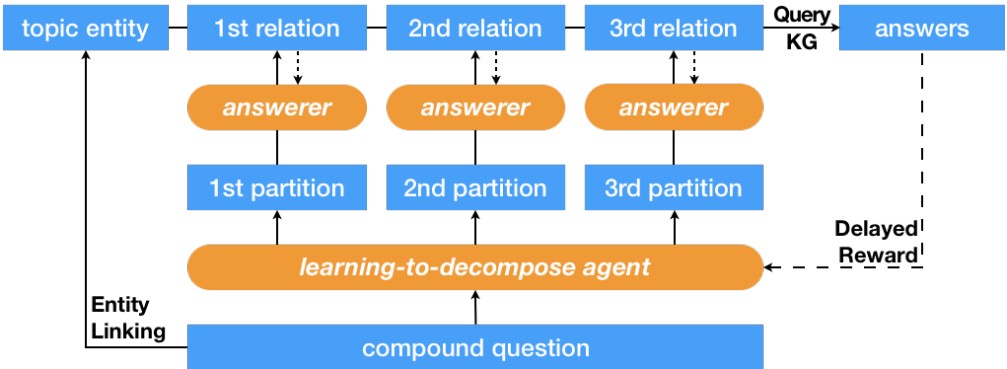

Figure 2: An overview of our model and the flow of data. Two orange rounded rectangles correspond to components of our model, a learning-to-decompose agent and three simple-question answerers. The blue rectangles represent data in different forms. The solid line indicates the process of transforming data points, while the dashed line indicates the feedback (loss or reward) received by our model.

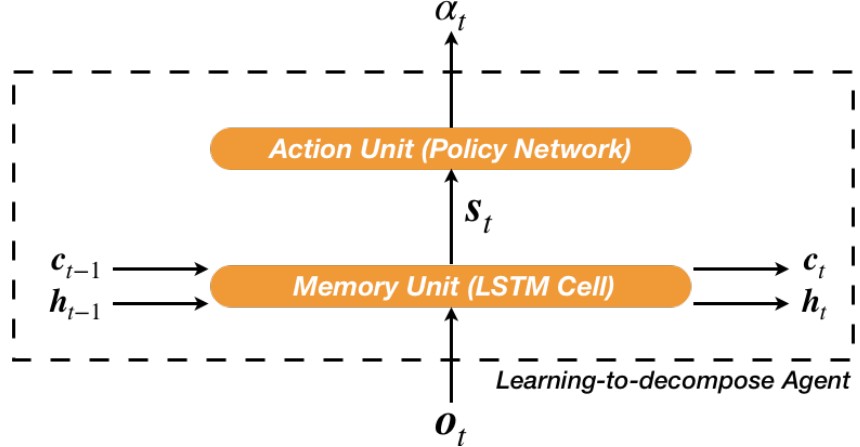

Figure 3: A zoom-in version of the lower half of figure 2. Our agent consists of two components: a *Memory Unit* and an *Action Unit*. The Memory Unit observes current word at each time step $t$ and updates the state of its own memory. We use a feedforward neural network as policy network for the Action Unit.

question of $N$ words[3] $\mathbf{x} = \{\mathbf{x}_1, \mathbf{x}_2, \ldots, \mathbf{x}_N\}$, we assume that a sequence of words is essentially a partially observable environment and we can only observe the corresponding vector representation $\mathbf{o}_t = \boldsymbol{x}_t \in \mathbb{R}^D$ at time step $t$. Figure 3 summarizes the process for generating decision of compound question decomposition.

**Memory Unit** The agent has a Long Short-Term Memory (LSTM; Hochreiter & Schmidhuber (1997)) cell unrolling for each time step to memorize input history.

$$
\begin{aligned}
\boldsymbol{i}_t &= \sigma(\boldsymbol{W}_i[\boldsymbol{x}_t, \boldsymbol{h}_{t-1}] + \boldsymbol{b}_i) & \boldsymbol{f}_t &= \sigma(\boldsymbol{W}_f[\boldsymbol{x}_t, \boldsymbol{h}_{t-1}] + \boldsymbol{b}_f) \\
\boldsymbol{g}_t &= \tanh(\boldsymbol{W}_g[\boldsymbol{x}_t, \boldsymbol{h}_{t-1}] + \boldsymbol{b}_g) & \boldsymbol{o}_t &= \sigma(\boldsymbol{W}_o[\boldsymbol{x}_t, \boldsymbol{h}_{t-1}] + \boldsymbol{b}_o) \\
\boldsymbol{c}_t &= \boldsymbol{f}_t \odot \boldsymbol{c}_{t-1} + \boldsymbol{i}_t \odot \boldsymbol{g}_t & \boldsymbol{h}_t &= \boldsymbol{o}_t \odot \tanh(\boldsymbol{c}_t)
\end{aligned}
\tag{1}
$$

where $\boldsymbol{W}_i, \boldsymbol{W}_f, \boldsymbol{W}_g, \boldsymbol{W}_o \in \mathbb{R}^{H \times (D+H)}$, $\boldsymbol{b}_i, \boldsymbol{b}_f, \boldsymbol{b}_g, \boldsymbol{b}_o \in \mathbb{R}^H$, and $[\cdot, \cdot]$ denotes the concatenation of two vectors. $\sigma(\cdot)$ is the element-wise sigmoid activation function.

---

[3]Note that the length of sequences may vary from questions to questions. We handle such dynamic directly without padding zeros. The similar situation exists in a game environment that allows early stopping.

The state $s_t \in \mathbb{R}^{2H}$ of the agent is defined as

$$s_t = [c_t, h_t] \tag{2}$$

which maintained by the above memory cell (Eq. 1) unrolling for each time step. $[\cdot, \cdot]$ denotes the concatenation of two vectors.

**Action Unit** The agent also has a stochastic policy network $\pi(\alpha|s; W_\pi)$ where $W_\pi$ is the parameter of the network. Specifically, we use a two-layer feedforward network that takes the agent's state $s$ as its input:

$$\pi(\alpha|s; W_\pi) \propto \exp(W_\pi^{(2)}(\text{ReLU}(W_\pi^{(1)}s + b_\pi^{(1)})) + b_\pi^{(2)}) \tag{3}$$

where $W_\pi^{(1)} \in \mathbb{R}^{H \times 2H}, b_\pi^{(1)} \in \mathbb{R}^H, W_\pi^{(2)} \in \mathbb{R}^{3 \times H}$ and $b_\pi^{(2)} \in \mathbb{R}^3$.

Following the learned policy, the agent decomposes a question of length $N$ by generating a sequence of actions $\alpha_t \in \{1st, 2nd, 3rd\}, t = 1, 2, \ldots, N$. Words under the same decision (e.g. *1st*) will be appended into the same sub-sequence (e.g. *the first partition*).

Formally, $\mathbf{x}^{(k)} = \{\mathbf{x}_1^{(k)}, \mathbf{x}_2^{(k)}, \ldots, \mathbf{x}_{t_k}^{(k)}\}, k = 1, 2, 3$ denotes the partitions of a question. Note that in a partition, words are not necessarily consecutive[4]. The relative position of two words in original question is preserved. $t_1 + t_2 + t_3 = N$ holds for every question.

**Reward** The episodic reward $R$ will be $+1$ if the agent helps all the answerers to get the golden answers after each episode, or $-1$ otherwise. There is another reward function $R = \Sigma \log P(Y^* \mid X)$ that is widely used in the literature of using reinforcement learning for natural language processing task (Bahdanau et al., 2017; Zhang et al., 2018). We choose the former as reward function for lower variance.

Each unique rollout (sequence of actions) corresponds to unique compound question decomposition. We do *not* assume that any question should be divided into exactly three parts. We allow our agent to explore the search space of partition strategies and to increase the probability of good ones. The goal of our agent is to learn partition strategies that benefits the answerers the most.

## 3.2 SIMPLE-QUESTION ANSWERERS

With the help of the learning-to-decompose agent, simple-question answerers can answer compound questions. Once the question is decomposed into partitions as simple questions, each answerer takes its partition $\mathbf{x}^{(k)} = \{\mathbf{x}_1^{(k)}, \mathbf{x}_2^{(k)}, \ldots, \mathbf{x}_{t_k}^{(k)}\}$ as input and classifies it as the corresponding relation in knowledge graph.

For each partition $\mathbf{x}^{(k)}$, we use LSTM network to construct simple-question representation directly. The partition embedding is the last hidden state of LSTM network, denoted by $x^{(k)} \in \mathbb{R}^{2H}$. We again use a two-layer feedforward neural network to make prediction, i.e. estimate the likelihood of golden relation $r$.

$$P(\hat{y} = r \mid x^{(k)}; W_p) \propto \exp(W_p^{(2)}(\text{ReLU}(W_p^{(1)}x^{(k)} + b_p^{(1)})) + b_p^{(2)}) \tag{4}$$

where $W_p^{(1)} \in \mathbb{R}^{H \times 2H}, b_p^{(1)} \in \mathbb{R}^H, W_p^{(2)} \in \mathbb{R}^{3 \times H}$ and $b_p^{(2)} \in \mathbb{R}^C$. $C$ is the number of classes.

Each answerer only processes its corresponding partition and outputs a predicted relation. These three modules share no parameters except the embedding layer because our agent will generates conflict assignments for the same questions in different epoches. If all the answerers share same parameters in different layers, data conflicts undermine the decision boundary and leads to unstable training.

Note that we use a classification network for sequential inputs that is as simple as possible. In addition to facilitating the subsequent theoretical analysis, the simple-question answerers we proposed

---

[4]The subscripts of each partition are re-ordered for simplicity.

are much simpler than good baselines for simple question answering over knowledge graph, without modern architecture features such as bi-directional process, read-write memory (Bordes et al., 2015), attention mechanism (Yin et al., 2016) or residual connection (Yu et al., 2017).

The main reason is that our agent learns to decompose input compound questions to the simplified version which is answerable for such simple classifiers. This can be a strong evidence for validating the agent's ability on compound question decomposition.

### 3.3 Training our model

The agent and the answerers share the same embeddings. The agent can only observe word embeddings while the answerers are allowed to update them in the backward pass. We train three simple-question answerers separately using Cross Entropy loss between the predicted relation and the golden relation. These three answerers are independent of each other.

We do not use the pre-train trick for all the experiments since we have already observed consistent convergence on different task settings. We reduce the variance of Monte-Carlo Policy Gradient estimator by taking multiple ($\leq 5$) rollouts for each question and subtracting a baseline that estimates the expected future reward given the observation at each time step.

**The Baseline** We follow Ranzato et al. (2016) which uses a linear regressor which takes the agent's memory state $s_t$ as input and minimizes the mean squared loss for training. Such a loss signal is used for updating the parameters of baseline only. The regressor is an unbiased estimator of expected future rewards since it only depends on the agent's memory states.

Our agent learns a optimal policy to decompose compound questions into simple ones using Monte-Carlo Policy Gradient (MCPG) method. The partitions of question is then fed to corresponding simple-question answerers for policy evaluation. The agent takes the final episodic reward in return.

## 4 Experiments

The goal of our experiments is to evaluate our hypothesis that our model discovers useful question partitions and composition orders that benefit simple-question answerers to tackle compound question answering. Our experiments are three-fold. First, we trained the proposed model to master the order of arithmetic operators (e.g., $+ - \times \div$) on an artificial dataset. Second, we evaluate our method on the standard benchmark dataset **MetaQA** (Zhang et al., 2017). Finally, we discuss some interesting properties of our agent by case study.

### 4.1 Mastering Arithmetic Skills

The agent's ability of compound question decomposition can be viewed as the ability of priority assignment. To validate the decomposition ability of our proposed model, we train our model to master the order of arithmetic operations. We generate an artificial dataset of complex algebraic expressions. (e.g. $1 + 2 - 3 \times 4 \div 5 =$? or $1 + (2 - 3) \times 4 \div 5$). The algebraic expression is essentially a question in math language which the corresponding answer is simply a real number.

Specifically, the complex algebraic expression is a sequence of arithmetic operators including $+$, $-$, $\times$, $\div$, ( and ). We randomly sample a symbol sequence of length $N$, with restriction of the legality of parentheses. The number of parentheses is $P(\leq 2)$. The number of symbols surrounding by parentheses is $Q$. The position of parentheses is randomly selected to increase the diversity of expression patterns. For example, $(+\times)+(\div)$ and $+\times(+\times)-\div$ are data points $(1+2\times3)+(4\div5)$ and $1 + 2 \times (3 + 4 \times 5) - 6 \div 7$ with $N = 8$.

This task aims to test the learning-to-decompose agent whether it can assign a feasible order of arithmetic operations. We require the agent to assign higher priority for operations surrounding by parentheses and lower priority for the rest of operations. We also require that our agent can learn a policy from short expressions ($N \leq 8$), which generalizes to long ones ($13 \leq N \leq 16$).

We use 100-dimensional ($D = 100$) embeddings for symbols with Glorot initialization (Glorot & Bengio, 2010). The dimension of hidden state and cell state of memory unit H is 128. We use the RMSProp optimizer (Tieleman & Hinton, 2012) to train all the networks with the parameters

recommended in the original paper except the learning rate $\alpha$. The learning rate for the agent and the answerers is 0.00001 while the learning rate for the baseline is 0.0001. We test the performance in different settings. Table 1 summarizes the experiment results.

Table 1: Agent Performance under Different Settings.

| Train | Test | Test ACC |
|---|---|---|
| $N = 5, P = 0, Q = 0$ | $N = 20, P = 0, Q = 0$ | 99.21 |
| $N = 8, P = 1, Q = 3$ | $N = 13, P = 1, Q = 3$ | 93.37 |
| $N = 8, P = 1, Q = 3$ | $N = 13, P = 1, Q = 7$ | 66.42 |

The first line indicates that our agent learns an arithmetic skill that multiplication and division have higher priority than addition and subtraction. The second line indicates that our agent learns to discover the higher-priority expression between parentheses. The third line, compared to the second line, indicates that increasing the distance between two parentheses could harm the performance. We argue that this is because of the Long Short-Term Memory Unit of our agent suffers when carrying the information of left parenthesis for such a long distance.

## 4.2 KBQA BENCHMARK

We evaluate our proposed model on the test set of two challenging KBQA research dataset, i.e., WebQuestions (Berant et al., 2013) and MetaQA (Zhang et al., 2017). Each question in both datasets is labeled with the golden topic entity and the inference chain of relations. The statistics of MetaQA dataset is shown in table 2. The number of compound questions in MetaQA is roughly twice that of simple questions. The max length of training questions is 16. The size of vocabulary in questions is 39,568.

Table 2: Data Statistics on MetaQA Dataset.

|  | 1-hop | 2-hop | 3-hop | Total |
|---|---|---|---|---|
| Train | 96,106 | 118,980 | 114,196 | 329,282 |
| Dev | 9,992 | 14,872 | 14,274 | 39,138 |
| Test | 9,947 | 14,872 | 14,274 | 39,093 |

The coupled knowledge graph contains 43,234 entities and 9 relations. We also augmented the relation set with the inversed relations, as well as a "NO_OP" relation as placeholder. The total number of relations we used is 14 since some inversed relations are meaningless.

WebQuestions contains 2,834 questions for training, 944 questions for validation and 2,032 questions for testing respectively. We use 602 relations for the relation classification task. The number of compound questions in WebQuestions is roughly equal to that of simple questions. Note that a compound question in WebQuestions is decomposed into two partitions since the maximum number of corresponding relations is two.

One can either assume topic entity of each question is linked or use a simple string matching heuristic like character trigrams matching to link topic entity to knowledge graph directly. We use the former setting while the performance of the latter is reasonable good. We tend to evaluate the relation detection performance directly.

For both datasets, we use 100-dimensional ($D = 100$) word embeddings with Glorot initialization (Glorot & Bengio, 2010). The dimension of hidden state and cell state of memory unit $H$ is 128. We use the RMSProp optimizer (Tieleman & Hinton, 2012) to train the agent with the parameters recommended in the original paper except the learning rate. We train the rest of our model using Adam (Kingma & Ba, 2015) with default parameters. The learning rate for all the modules is 0.0001 no matter the optimizer it is. We use four samples for Monte-Carlo Policy Gradient estimator of REINFORCE. The metric for relation detection is overall accuracy that only cumulates itself if all relations of a compound question are correct.

Table 3 presents our results on MetaQA dataset. The last column for total accuracy is the most representative for our model's performance since the only assumption we made about input questions is the number of corresponding relations is at most three.

Table 3: Accuracy on MetaQA test set.

| Model | 1-hop | 2-hop | 3-hop | Total |
|---|---|---|---|---|
| KV-MemNN | 95.8 | 25.1 | 10.1 | 37.6 |
| Bordes, Chopra, and Weston's QA system | 95.7 | 81.8 | 28.4 | 65.8 |
| VRN (Zhang et al., 2017) | 97.5 | 89.9 | 62.5 | 81.8 |
| Ours | **98.1** | **90.8** | **83.4** | **90.0** |

Table 4: Accuracy on WebQuestions relation detection.

| Model | Accuracy |
|---|---|
| HR-BiLSTM (Yu et al., 2017) | 82.53 |
| HR-BiLSTM w/o relation_name | 81.69 |
| HR-BiLSTM w/o relation_words | 79.68 |
| Ours | **81.74** |

Table 4 presents our results on WebQuestions Dataset. Note that there are $5\%$ of relations in test set that have never seen in the training samples. To address this issue, recent advances (Yu et al., 2017) on this dataset focus on leveraging information of the name of Freebase relation while we are only using question information for classification.

## 4.3 ABLATION STUDY

We assume that the compound question can be decomposed into at most three simple questions. In practice, this generalized assumption of answerable questions is not necessary. One example is that WebQuestions only contains compound questions corresponding to two but not three relations. It indicates that people tend to ask less complicated questions more often. So we conduct an ablation study for the hyperparameters of this central assumption in our paper.

We assume that all the questions in MetaQA dataset contain at most two corresponding relations. We run the same code with the same hyperparameters except we only use two simple-question answerers. The purpose of the evaluation is to prove that our model improves performance on 1-hop and 2-hop questions by giving up the ability to answer three-hop questions. Table 5 presents our results on ablation test. We can draw a conclusion that there exists a trade-off between answering more complex questions and achieving better performance by limiting the size of search space.

## 4.4 CASE STUDY

Figure 4 illustrates a continuous example of figure 1 for the case study, which is generated by our learning-to-decompose agent. Assuming the topic entity $e$ is detected and replaced by a place-holder, the agent may discover two different structures of the question that is consistent with human intuition. Since the knowledge graph does not have a movie-to-movie relation named *"share_actor_with"*, the lower partition can not help the answerers classify relations correctly. However, the upper partition will be rewarded. As a result, our agent optimizes its strategies such that it can decompose the original question in the way that benefits the downstream answerers the most.

We observe the fact that our model understands the concept of "share" as the behavior "take the inversed relation". That is, "share actors" in a question is decomposed to "share" and "actors" in two partitions. The corresponding formulation is $g(e) = f_2(f_1(e)) = (f_2 \circ f_1)(e)$. We observe the same phenomenon on "share directors". We believe it is a set of strong evidence for supporting our main claims.

## 5 DISCUSSION

Understanding compound questions, in terms of The Principle of Semantic Compositionality (Pelletier, 1994), require one to decompose the meaning of a whole into the meaning of parts. While previous works focus on leveraging knowledge graph for generating a feasible path to answers, we

Table 5: Ablation study for accuracy on MetaQA test set.

| Model | 1-hop | 2-hop | Total |
|---|---|---|---|
| Ours (Three Answerers) | 98.1 | 90.8 | 93.7 |
| Ours (Two Answerers) | **99.2** | **93.1** | **95.5** |

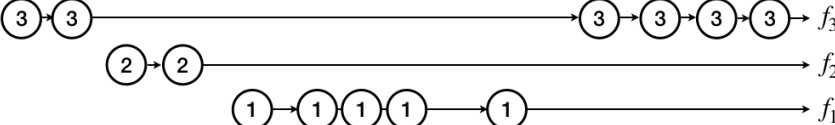

the films that share actors with the film *[Topic_Entity]* are written by who

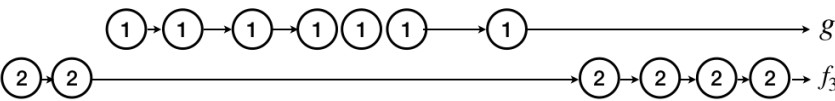

Figure 4: A continuous example of figure 1. The hollow circle indicates the corresponding action the agent takes for each time step. The upper half is the actual prediction while the lower half is a potential partition. Since we do not allow a word to join two partitions, the agent learns to separate "share" and "actors" into different partitions to maximize information utilization.

propose a novel approach making full use of question semantics efficiently, in terms of the Principle of Semantic Compositionality.

In other words, it is counterintuitive that compressing the whole meaning of a variable-length sentence to a fixed-length vector, which leaves the burden to the downstream relation classifier. In contrast, we assume that a compound question can be decomposed into three simple questions at most. Our model generates partitions by a learned policy given a question. The vector representations of each partition are then fed into the downstream relation classifier.

While previous works focus on leveraging knowledge graph for generating a feasible path to answers, we propose a novel approach making full use of question semantics efficiently, in terms of the Principle of Semantic Compositionality.

Our learning-to-decompose agent can also serve as a plug-and-play module for other question answering task that requires to understand compound questions. This paper is an example of how to help the simple-question answerers to understand compound questions. The answerable question assumption must be relaxed in order to generalize question answering.

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
