# OpenReview forum: "Learning to Decompose Compound Questions with Reinforcement Learning"
_ICLR.cc/2019/Conference_

### Official Review · AnonReviewer3 · 2018-10-31
**Lack of comparison with previous state-of-the-art methods over more widely used benchmarks**

**Rating:** 5
**Confidence:** 4

**Review:**

This paper proposes a new approach for answering questions requiring multi-hop reasoning. The key idea is to introduce a sequence labeler to divide the question into at most 3 parts, each part corresponds to a relation-tuple. The labeler is trained with the whole KB-QA pipeline with REINFORCE in an end-to-end way.

The proposed approach was applied to a synthetic dataset and a new KB-QA dataset MetaQA, and achieves good results.

I like the proposed idea, which sounds a straightforward solution to compound question answering. I also like the clarification between "compound questions" instead of "multi-hop questions". In my opinion, "multi-hop questions" can also refer to the cases where the questions (can be simple questions) require multi-hop over evidence to answer.

My only concern is about the evaluation on MetaQA, which seems a not widely used dataset in our community. Therefore I am wondering whether the authors could address the following related questions in the rebuttal or revision:

(1) I was surprised that WebQuestions is not used in the experiments. Could you explain the reason? My guess is that WebQuestions contains compound questions that cannot be simply decomposed as sequence labeling, because that some parts of the question can participant in different relations. If this is not true, could you provide results on WebQuestions (or WebQSP).

(2) There were several previous methods proposed for decomposition of compound questions, although they are not proposed for KB-QA. Examples include "Search-based Neural Structured Learning for Sequential Question Answering" and "ComplexWebQuestions". I think the authors should compare their approach with previous work. One choice is to reimplement their methods. An easier option might be applying the proposed methods to some previous datasets, because the proposed method is not specific to KB-QA, as long as the simple question answerer is replaced to other components like a reader in the ComplexWebQuestions work.

---

> ### Author Response · Authors · 2018-11-27
> **Thank you very much for your insightful review! Paper updates and model improves!**
>
> Thank you very much for your insightful review! We have updated our paper with new experiments! We will address your concerns point by point.
>
> Please refer to global comments for brief version of model improvement and paper refinement!
>
> Q1: Could you provide results on WebQuestions (or WebQSP).
> A1: Yes! We conduct experiments on WebQuestions relation detection since relation detection is believed to be the bottleneck of KBQA and we attempt to solve it. It achieves competitive results to strong baseline (Yu et al, 2017, [1]).
>
> It seems like our model performs differently in two datasets. Here are the reasons:
> - There are ~5% relations that remains unseen in training set. It is a harmful setting for classification task.
> - To address the above issue, recent approaches try to leverage the information from knowledge base, especially the detailed name or schema info of Freebase relations. By contrast, our proposed model only leverages the question information to achieve competitive results.
>
> Q2: "I think the authors should compare their approach with previous work."
> A2: We have updated our paper for discussion in related work (Please check out paragraph 2 & 3 in section 2.2). We tried to reimplement their methods and found that it is not suitable for our setup.
>
> * Search-based Neural Structured Learning for Sequential Question Answering
> When generating datasets, the author employs crowdsourcing workers to manually decompose questions from WikiTableQuestions into sequential questions. It aims to train a text-to-sql model for querying answers and updating next input question interactively. Conversely, our proposed model emphasizes decomposing compound questions automatically with fewer supervision.
>
> * ComplexWebQuestions
> - The state-of-the-art solution of ComplexWebQuestions adopts pointer network to decompose complex web questions into simple ones. This decomposition process is guided by supervisions inline with human logic (e.g., conjunction or composition etc.). The author feeds all the decomposed questions into search engine then collects top-ranked web snippets as data source of answers.
>
> - Note that the pointer network is trained via maximizing log-likelihood of annotations.
>
> - The problem is that, if we replace pointer network with our learning-to-decompose agent, we cannot afford to crawl web pages during training because our agent will generate different partitions.
>
> Thank you again for your valuable review and inspiration! We would be happy to open source our code and hyper-parameters until the final decisions are out!
>
> [1] Yu et al. Improve Neural Relation Detection for Knowledge-based Question Answering. ACL, 2017.

---

### Official Review · AnonReviewer1 · 2018-11-02
**Interesting idea. Lacking technical details and error analysis.**

**Rating:** 5
**Confidence:** 3

**Review:**

This paper proposes a knowledge-based QA system that learns to decompose compound questions into simple ones. The decomposition is modeled by assigning each token in the input question to one of the partitions and receiving reward signal based on the final gold answer. The model achieves the state-of-the-art performance on the MetaQA dataset.

My main complaint about the paper is its lack of technical details and analysis of empirical results. Parts of the paper seem quite unclear, for example:

In the last paragraph of Section 3.1, it says “We do not assume that nay question should be divided into exactly three parts. … See section 4 for case study.” Does this mean that the model can have <=3 partitions, but not more? How is this number decided?

Section 3.2 describes the simple-question answer. From Eq (4), it seems that the answerer only uses the current partition, is that the case? Moreover, how is the gold relation r obtained?

It would be nice to add more explanation to the caption of Figure 4 to make it self-contained.

The case study section (4.3) only contains a single example. It would be very helpful to include more examples of question partitions (there is enough space). Error analysis would also be helpful to understand, for example, why the proposed model is worse than VRN (Zhang et al. 2017) on 1- and 2-hop questions.

---

> ### Author Response · Authors · 2018-11-27
> **Thank you very much for your helpful reviews! We have updated our paper for clarification.**
>
> Thank you very much for your valuable review! We have updated our paper with additional experiments! We will provide detailed explanation for your concerns.
>
> Please refer to global comments for brief version of model improvement and paper refinement!
>
> Q1: Does this mean that the model can have <=3 partitions, but not more? How is this number decided?
> A1:
> - The central assumption of our paper is to generalize the assumption of answerable questions from simple questions to compound questions (simple questions included).
>
> - Based on the observation of daily questions asked by people (e.g. WebQuestions) and the currently available datasets (MetaQA), it is hard to find compound questions with more than three partitions to experiment with. So the default number of partitions is 2 or 3 (<=3). We have updated our paper for ablation test of these two options. Results and discussion can be found in section 4.3.
>
> Q2: From Eq (4), it seems that the answerer only uses the current partition, is that the case? Moreover, how is the gold relation r obtained?
> A2: In our improved model, we use three answerers for each partition. The vector representation of a partition is the last hidden state of answerer's LSTM network. The golden relation $r$ is part of the golden label providing by datasets. The answerer predicts and updates according to the gradients of cross entropy loss.
>
> Q3: It would be nice to add more explanation to the caption of Figure 4 to make it self-contained.
> A3: We have updated our paper to make it self-contained! Please check out our paper for more details.
>
> Q4: The case study section (4.3) only contains a single example. It would be very helpful to include more examples of question partitions (there is enough space). Error analysis would also be helpful to understand, for example, why the proposed model is worse than VRN (Zhang et al. 2017) on 1- and 2-hop questions.
> A4:
> - Case Study is now section 4.4! The main purpose of case study is to illustrate that our agent can maximize information utilization by assigning words to the best position.
>
> - We also add an ablation test for providing better understanding of our model. Since we have further improved our model, please refer to global comments for reasons of model change. It directly leads to outperforming the state-of-the-art model by ~8% overall accuracy.
>
> Thank you again for your time and helpful review! We really appreciate it! We would be happy to open source our code and hyper-parameters until the final decisions are out!

---

### Official Review · AnonReviewer2 · 2018-11-05
**Good paper, but need more related work discussions**

**Rating:** 6
**Confidence:** 5

**Review:**

Summary: the paper is interested in parsing compound questions for querying on knowledge graph, e.g. MetaQA by Zhang et al. (2017). The paper proposes to have two modules, one that segments the question into partitions (up to three) and the other that looks at each segment to get the relation. The relations are merged to obtain a single KG path, which is queried to obtain the answer. Since the segmentation is a non-differentiable process, the paper uses reinforcement learning to propagate gradient to the segmentation model. The segmentation is a process of classifying each word for which partition it should be tied to. Answering is a process of classifying the partition into one of the possible relation edges. The model shows expected results in a synthetic arithmetic dataset, and obtains the state of the art in MetaQA, improving nearly 5% over the baseline. The model especially does much better on 3-hop questions, with nearly 20% improvement.

Strengths: the paper is well-written. The model is simple yet effective and is a novel contribution to compound question answering on KG. Especially, the improvement on 3-hop category is nearly 20%, which is substantial and quite impressive.

Weaknesses: My biggest concern is the lack of discussions on its relevance to  (Iyyer et al., 2016), which also proposed to decompose question into simpler ones for WIkiTableQuestions. Also, I think it would be good to mention Semantic Role Labeling as related literature, which is about tagging each word with its role in the sentence. The partition index can be somewhat considered as a “role” in the sentence.

Questions:
1. How do you obtain x^(k)? Is it the last state of the LSTM?
2. Why did you have to augment “NO_OP” relation in the MetaQA dataset?
3. Why +1 reward has lower variance than probabilistic reward? Explanation or citation would be needed.
4. What if two partitions need to share a word? The current setup necessitates that a word participates in only one partition. Wouldn’t this be problematic?
5. I am a bit confused about how the simple question answering module is trained. Is it directly trained by the gold relation label?

Typos and Suggestions:
- Second paragraph of 2.1: in stead -> instead
- Third paragraph of 2.1: research. -> research
- c_t + h_t: would be good to explicitly mention that the circled plus sign is concatenation.
- Last paragraph on page 4: “leave to be”?
- Second last paragraph of 4.1: he -> The
- Second paragraph of 4.2: “if exists a proper meaning”?
- First paragraph of page 7: be either assume -> either assume
- Last paragraph of Section 5: generalizing -> generalize
- I think you should not put acknowledgment in a double-blind submission.

M Iyyer, W Yih, MW Chang. Answering complicated question intents expressed in decomposed question sequences. 2016 (https://arxiv.org/abs/1611.01242)

---

> ### Author Response · Authors · 2018-11-27
> **Thank you very much! Please check out our latest version of paper for model improvement and paper refinement!**
>
> Thank you very much for your detailed and helpful review! We have updated our paper with your suggestions! We will address your concerns point by point. Please refer to global comments for brief version of model improvement and paper refinement!
>
> * Reply for Weakness
> We compare [1] with our work and summarize an important line of Semantic Role Labeling in our latest paper. We like to point out that [1] decomposes WikiTableQuestions into sequential questions by crowdsourcing workers (manually) in the process of generating SequentialQA dataset. However, we train our agent to learn to decompose questions automatically.
>
> Semantic Role Labeling is similar to labeling priority/actions word by word, which is part of our proposed method. However, we don't require supervision signals at the token-level. The only supervision for our agent is the +1/-1 reward as feedbacks.
>
> * Questions
> Q1: How do you obtain x^(k)? Is it the last state of the LSTM?
> A1: Yes, it is the last hidden state $h$ of the LSTM. For clarity, there are two x^(k) with different style in our paper. The bold x^(k) denotes a sub-sequence of words as a partition. Another bold italic x^(k) denotes the final vector representation of corresponding partition. We have updated our paper for clarification.
>
> Q2: Why did you have to augment “NO_OP” relation in the MetaQA dataset?
> A2:
> (1) The main reason for augmenting a dummy relation is to provide more freedom of the cooperation among our agent and the answerers\*. When our model is trying to answer a simple question, our agent may filter some unrelated words (e.g. stop words) out of the first partition because of its stochastic policy. The second and the third answerer can return a "NO_OP" relation when receiving some meaningless inputs.
>
> (2) If we don't augment a "NO_OP" relation, our agent has to assign every single word to the first partition and hope the first answerer can predict the golden relation correctly, which is too strict for a feasible solution. Note that we allow our agent to learn partition strategies that is different from human intuition since we train it using RL settings.
>
> Q3: Why +1 reward has lower variance than probabilistic reward? Explanation or citation would be needed.
> A3:
> (1) Because the maximum value of variance of +1/-1 reward is 1 (Please see proof below). The variance of probabilistic reward does not necessarily have an upper bound since the value of logarithmic function goes negative infinity if the likelihood is sufficiently small. This kind of situation is likely to occur in the early stages of training when the agent explores the space of partition strategies actively.
>
> (2) From the perspective of model design, we have tried our best to disentangle our model, i.e. prohibiting our agent to update the embedding layer and use +1/-1 reward. If the agent is allowed to observe probabilistic reward as feedback, it will greedily maximize partial reward (say the first term of the sum of log-likelihood). The feedback from first answerer will dominate before the agent fully explores the search space. Hence the model is likely to collapse which leads to unstable training.
>
> [Proof]: Suppose the probability mass function (PMF) of reward is defined as
> $p_X(x) = p if x = +1
>               = q if x = -1, p + q = 1.$
> The expected reward is $E[x] = p + (1 - p)(-1) = 2p - 1$.
> The variance is
> 	$Var[x] = \Sigma (x - E[x])^2 \times p(x)
> 	              = 4pq
> 	            <= 4 ((p + q) / 2)^2
> 	              = 1$,
> with equality if and only if $p = q = 0.5$. #
>
> Q4: What if two partitions need to share a word? The current setup necessitates that a word participates in only one partition. Wouldn’t this be problematic?
> A4: No, we provide the following four explanations.
> (1) Maximum Information Utilization. The current setup forces the agent to fully explore the search space of partition strategies such that each word in the questions contributes to the confidence of downstream classifiers. Imagine a key word being misplaced, one classifier losses information and the other classifier receives extra noise, which harms information utilization significantly.
>
> (2) Performance and size of search space tradeoff. If we allow two partitions to share a word, the size of search space increases by a factor of 2^N (from 3^N to 6^N). N denotes the length of a question. It would be interesting to further investigate whether the generalized model is able to converge and produce better results or interpretability.
>
> Q5: I am a bit confused about how the simple question answering module is trained. Is it directly trained by the gold relation label?
> A5: Sorry for the confusion. Yes. It is fair to train simple question answerers by the gold relation label directly, compared to training three independent question classifier using the same supervision. We have updated our paper for clarification.

---

### Author Response · Authors · 2018-11-27
**Dear reviewers, we thread a global comment for improvement on both our model and paper.**

## Main Improvement
1. We have improved our model by replacing the answerer into three identical simple-question answerers with embedding layer shared.

* Reasons
During experiments, we observe that if we share the simple-question answerer across different partitions of a question, our agent may generate conflict assignments at the beginning of training process. Data conflicts undermine the decision boundary learned by the answerer (classifier).

* An Example
 Considering a four-word question "w1 w2 w3 w4?", the agent generates two different labeling "1st 1st 2nd 2nd" and "2nd 2nd 1st 1st" in two different epoch. Since the answerer is shared,

- the former mapping (f("w1 w2") -> 1st_golden_relation) and
- the latter mapping    (f("w1 w2") -> 2nd_golden_relation) is conflicting.

* Performance Improvement
- On MetaQA, our model now outperforms state-of-the-art by ~8% overall accuracy, and
- On WebQuestions, our model achieves competitive result to results that leverage knowledge base information by only using question information.
- Details are described in section 4.2 of our paper.

2. We have updated the following subsections.
- Section 2.2 for detailed comparison of Iyyer et al., 2016 [1] and other works related to complex questions;
- Section 2.3 for Deep Semantic Role Labeling and its relevance to our work;
- Section 3.2 for describing our improved model architecture;
- Section 3.3 for more training details;
- Section 4.2 for benchmarking WebQuestions which is widely used in the KBQA community;
- Section 4.3 for ablation study to conclude that there exists a tradeoff between model assumption and model performance.

---

### Meta-Review · Area_Chair1 · 2018-12-17
**interesting directions / results are not very convincing**

**Confidence:** 4
**Recommendation:** Reject

**Metareview:**

+ an interesting task -- learning to decompose questions without supervision

- reviewers are not convinced by evaluation. Initially evaluated on MetaQA only, later relation classification on WebQuestions has been added.  It is not really clear that the approach is indeed beneficial on WebQuestion relation classification (no analysis / ablations) and MetaQA is not a very standard dataset.

-  Reviewers have concerns about comparison to previous work / the lack of state-of-the-art baselines. Some of these issues have been addressed though (e.g., discussion of Iyyer et al. 2016)